# Nutrigenomic Effects of White Rice and Brown Rice on the Pathogenesis of Metabolic Disorders in a Fruit Fly Model

**DOI:** 10.3390/molecules28020532

**Published:** 2023-01-05

**Authors:** Saheed Olanrewaju Saka, Yusuf Yahaya Salisu, Hauwa’u Muhammad Sahabi, Kamaldeen Olalekan Sanusi, Kasimu Ghandi Ibrahim, Murtala Bello Abubakar, Suleiman Ahmed Isa, Muhammad Gidado Liman, Sha’aya’u Shehu, Ibrahim Malami, Kim Wei Chan, Nur Hanisah Azmi, Mustapha Umar Imam

**Affiliations:** 1Centre for Advanced Medical Research and Training, Usmanu Danfodiyo University, Sokoto P.M.B. 2254, Nigeria; 2Department of Physiology, Faculty of Basic Medical Sciences, College of Health Sciences, Usmanu Danfodiyo University, Sokoto P.M.B. 2346, Nigeria; 3Department of Biochemistry and Molecular Biology, Faculty of Science, Usmanu Danfodiyo University, Sokoto P.M.B. 2346, Nigeria; 4Department of Basic Medical and Dental Sciences, Faculty of Dentistry, Zarqa University, Zarqa 13110, Jordan; 5Department of Pure and Applied Chemistry, Faculty of Science, Usmanu Danfodiyo University, Sokoto P.M.B. 2346, Nigeria; 6Department of Pharmacognosy and Ethnopharmacy, Faculty of Pharmaceutical Sciences, Usmanu Danfodiyo University, Sokoto P.M.B. 2346, Nigeria; 7Natural Medicines and Products Research Laboratory, Institute of Bioscience, Universiti Putra Malaysia, UPM, Serdang 43400, Selangor, Malaysia; 8Faculty of Food Science and Nutrition, Universiti Malaysia Sabah, Kota Kinabalu 88400, Sabah, Malaysia; 9Department of Medical Biochemistry, Faculty of Basic Medical Sciences, College of Health Sciences, Usmanu Danfodiyo University, Sokoto P.M.B. 2346, Nigeria

**Keywords:** brown rice, white rice, metabolic syndrome, insulin resistance, *dPEPCK*, *dIRS*, *dACC*, oxidative stress, obesity, fruit fly

## Abstract

Consumption of white rice (WR) has been shown to predispose individuals to metabolic disorders. However, brown rice (BR), which is relatively richer in bioactive compounds, possesses anti-glycaemic and antioxidant effects. In this study, fifteen cultivars of paddy rice that are predominantly consumed in North West Nigeria were analysed for their nutritional composition, bioactive contents and effects on metabolic outcomes in a fruit fly model. Gene expression analyses were conducted on the whole fly, targeting *dPEPCK*, *dIRS*, and *dACC.* The protein, carbohydrate, and fibre contents and bioactives of all BR cultivars were significantly different (*p* < 0.05) from the WR cultivars. Moreover, it was demonstrated that the glucose and trehalose levels were significantly higher (*p* < 0.05), while glycogen was significantly lower (*p* < 0.05) in the WR groups compared to the BR groups. Similarly, the expression of *dACC* and *dPEPCK* was upregulated, while that of *dIRS* was downregulated in the WR groups compared to the BR groups. Sex differences (*p* < 0.05) were observed in the WR groups in relation to the nutrigenomic effects. Our findings confirm metabolic perturbations in fruit flies following consumption of WR via distortion of insulin signalling and activation of glycogenolysis and gluconeogenesis. BR prevented these metabolic changes possibly due to its richer nutritional composition.

## 1. Introduction

Rice is a staple food among Nigeria’s population; its annual consumption is estimated to be around 5 million metric tonnes, the highest in sub-Saharan Africa [1]. Rice has been documented to exhibit a large variation in its composition from region to region due to inherent botanical differences resulting from its genetic properties, paddy soil on which the rice is grown, fertiliser application, climate, postharvest treatments and storage [2]. Brown rice (BR) is obtained following the de-husking of harvested paddy rice whereas white rice (WR) further undergoes polishing processes to remove the bran and germ portions of the intact grains, thereby losing several beneficial bioactive compounds, including oryzanols and phenolic compounds [3]. The availability and profile of these bioactive compounds vary across different BR varieties [4]. Due to the refining process, WR is composed mainly of carbohydrates [5] and has a starchy endosperm which is the most easily digestible carbohydrate, with a very high glycaemic index (GI) and glycaemic load (GL) [6,7].

WR is a staple food in many low- and middle-income countries (LMICs) facing rapid economic development and nutritional transition [8]. Consumption of WR as a staple food predisposes individuals to metabolic disorders, mostly as a result of loss of important bioactive compounds to the polishing processes [9,10,11,12]. Regular consumption of WR has been implicated in fat store expansion, which promotes higher weight gain, oxidized low density lipoprotein (ox-LDL) levels and worsening of lipid metabolism in rats as a result of a conversion of excess glucose to free fatty acids (FFAs) and triglycerides (TGs) [10,13]. This finding was supported by Abubakar et al. [14] who reported a 10% enhanced body weight in rats fed on low amylose WR compared with those on standard chow. Similarly, increased risk of body weight gain was associated with long-term consumption of WR after adjusting for confounding factors, which include obesogenic diets, sex, and age [15]. Similar studies on some rice cultivars such as MRQ74 (aromatic and high-quality rice variety), MRQ76 (fragrant paddy strain), MR219 (high-yielding mega-variety), and MR220 (local paddy) from Malaysia have shown that WR promotes the elevation of LDL, TG, and total cholesterol (TC), and decreases HDL levels compared to BR [10,14]. Conversely, BR obtained by de-husking of paddy rice has superior nutritional properties versus WR, although WR is the preferred choice among consumers because of its palatability. BR contains a variety of bioactive compounds that are abundantly located in the bran layer including oryzanols, phytosterols, tocotrienols, γ-aminobutyric acid (GABA), tocopherols, amino acids, acylated steryl glycoside (ASG), carotenoids, linoleic acid vitamins, minerals as well as functional proteins, unsaponifiable lipids and dietary fibre [16,17,18]. In addition, BR has abundant polyphenols which are among the functional components responsible for most of the nutraceutical values of the rice [19,20,21].

Owing to its genetic similarities with humans, the fruit fly (*Drosophila melanogaster*) is a versatile model used in the study of metabolic diseases [22,23]. Thus, this study sought to investigate the nutrigenomic effects of white rice and brown rice on the pathogenesis of metabolic disorders in *Drosophila melanogaster*.

## 2. Results

### 2.1. Proximate Analysis

From the analyses of the proximate composition, the protein content ranged from 10.91 ± 0.10% to 11.73 ± 0.10% among the BR cultivars. *Danboto*_B had the highest value of protein while *Yarwashagi*_B had the lowest value, whereas *Mai bakin-carki*_W (10.38 ± 0.18%) and *Yarkukuma*_W (2.43 ± 0.20%) had the highest and lowest values, respectively, among the WR cultivars. Similarly, the fibre content ranged from 2.33 ± 0.29% (*Yar kukuma*_B) to 5.33 ± 0.77% (*Akai mazahajji*_B and *Bakindanboto*_B) for BR, while it ranged from 0.17 ± 0.01% to 2.39 ± 0.27 for *Faro 44*_W and *Jamila*_W, respectively. The analyses showed that there was no significant difference (*p* > 0.05) in the protein and fibre compositions of all fifteen BR cultivars, but there were significant differences (*p* < 0.05) among the protein content of WR cultivars. The carbohydrate content ranged from 64.66 ± 0.67% to 71.08 ± 0.05% for the BR and from 62.34 ± 0.58% to 68.40 ± 0.93% for the WR. There were significant differences (*p* < 0.05) in the carbohydrate compositions of both the BR and WR cultivars except for *Babu Rashi*, *Dankaushi*, *Jeep* and *Baingila*. Moreover, there were no significant differences *(p* > 0.05) in the lipid content of all BR and WR cultivars studied, with the exception (*p* < 0.05) of *Danboto*_W, *Bakin danboto*_W, *Farro 44*_W, *Jirkita*_W and *Yarkukuma*_W. Similarly, no significant differences (*p* > 0.05) were observed in the moisture compositions of all the rice cultivars studied, except for (*p* < 0.05) both BR and WR cultivars of *Yarwashagi*, *Kwandala* and *Yarkatabore.* The ash contents had a higher value in the WR than in the BR. This ranged from 4.17 ± 0.32% to 7.83 ± 0.29% in BR and from 4.17 ± 0.26% to 9.50 ± 0.50% in WR. There were no significant differences (*p* > 0.05) between BR and WR cultivars, with the exception (*p* < 0.05) of *Baingila* cultivars (Table 1).

### 2.2. Bioactive Compounds Analyses

The TOC content ranged from 46.16 ± 0.05 to 48.15 ± 0.18 mg/DW of sample for BR and from 10.89 ± 0.09 to 21.29 ± 0.09 mg/DW of sample for WR. The analysis showed that there were significant differences (*p* < 0.05) between BR and WR of the same cultivar. The phenolic content ranged from 10.40 ± 0.20 to 11.43 ± 0.04 mg GAE/g DW for BR and from 0.18 ± 0.01 to 3.94 ± 0.00 mg GAE/g DW for WR. For the total flavonoid content, there were significant differences (*p* < 0.05) between all fifteen BR and WR (Table 2).

### 2.3. Effects on Body Weight Changes

Statistical analysis showed that there were significant differences (*p* < 0.05) between the BR, WR, and HFD groups for both male and female flies. However, there was no significant difference (*p* > 0.05) between WR cultivars in male groups, with the exception (*p* < 0.05) of JEEP, JAM, DANB, BAD, FARO, JIR and YARM (Table 3). The weight changes observed between BR groups and their corresponding WR groups were almost doubled among the male flies. Similarly, no significant difference (*p* > 0.05) was found between WR cultivars in female groups, with the exception (*p* < 0.05) of AKM, BAI, JAM and KWA. In addition, the male body weight was significantly higher than the female in all groups (Appendix A).

### 2.4. Effects on Negative Geotaxis

Less than eight flies were able to cross the 6 cm mark in both male and female flies of the WR and HFD groups. In contrast, about eight to ten fruit flies crossed the 6 cm mark in the NCD and BR groups. Analysis showed that there were significant differences (*p* < 0.05) between the BR, WR, and HFD groups for both male and female flies (Table 3). There was no significant difference (*p* > 0.05) between WR cultivars in both male and female groups, with the exception (*p* < 0.05) of BAD and FARO in the male group.

### 2.5. Effects on Biochemical Variables 

#### 2.5.1. Effects on Fasting Glucose Level

Both male and female fruit flies on BR had comparable fasting glucose levels to those of the NCD group. However, there were significant differences (*p* < 0.05) between the BR and WR groups for both male and female flies. Similarly, there was no significant difference (*p* > 0.05) between the WR groups of both male and female flies, except for (*p* < 0.05) YARK and DANB (Figure 1 and Figure 2) which were higher. In addition, the sex-dependent effect showed that the female groups in NCD, HFD, AKM, BAI, BAB, DAN, JEEP_B, JAM, MBC_W, YARW_B, KWA, YARK, DANB_W, and BAD had higher glucose levels compared with male the group (Appendix A).

#### 2.5.2. Effects on Trehalose Levels

The fruit flies on BR diets had comparable trehalose levels to those of the NCD group. There were significant differences *(p* < 0.05) between the BR and WR groups for both male and female flies. However, there was no significant difference *(p* > 0.05) between the WR groups of the male flies, except for JEEP, BAD, FARO, and YARM (Figure 1) which were higher *(p* < 0.05). Similarly, the WR of JEEP, BAD, FARO, JIR and YARM were significantly higher *(p* < 0.05) than other WR groups of female flies (Figure 2). In addition, no sex-dependent significant effect was observed in trehalose levels except in KWA_W and JIR_W, with higher levels in females (Appendix A).

#### 2.5.3. Effects on Glycogen Levels

The male groups on BR diets had comparable tissue glycogen levels to those of the NCD group, except for (*p* < 0.05) the AKM, MBC, YARW, DANB, and BAD groups which were higher (Figure 1). Similarly, there was no significant difference (*p* > 0.05) between BR groups of female flies, except for (*p* < 0.05) JEEP and DANB (Figure 2), which were higher. However, there were significant differences (*p* < 0.05) between the BR and WR groups for both male and female flies. In addition, sex-dependent significant differences were observed in AKM_B, JEEP_B, MBC, YARW_B, KWA_W, DANB_B, BAD, JIR_B, and YARM_B (Appendix A).

#### 2.5.4. Effects on Triglyceride Level

The male fruit flies on the BR diet had comparable triglyceride levels to the fruit flies on the NCD, except for JIR and YARM which are significantly higher (*p* < 0.05) (Figure 3). The BR groups were significantly different (*p* < 0.05) from their corresponding WR groups. Similarly, there were no significant differences in male fruit fly groups on WR except for the DANB, JIR, and YARM, which were significantly higher, and for the MBC which was significantly lower than the others *(p* < 0.05) (Figure 3). Among the female groups, DAN was significantly higher (*p* < 0.05) than other BR groups (Figure 4). Moreover, all the BR groups were significantly different (*p* < 0.05) from their corresponding WR groups. There was no significant difference (*p* > 0.05) between WR cultivars in female groups, with the exception (*p* < 0.05) of BAB and BAD (Figure 4), which were significantly higher. In addition, sex-dependent significant differences were observed in all groups except for MBC_B, BAD_B, FARO, JIR_B, and YARM (Appendix A).

#### 2.5.5. Effects on Oxidative Stress Markers 

The SOD activity for the male fruit flies of the BR groups was comparable to that of the NCD group, except for the BAI, BAB, JEEP, JAM, YARK, and YARM groups, which were significantly lower (*p* < 0.05) (Figure 3). Similarly, female fruit flies on BR diets had comparable SOD levels to the NCD group, except for the MBC groups, which were significantly different (*p* < 0.05) from them (Figure 4). All fruit flies on the WR diets had comparable SOD activity to the HFD group. In addition, sex-dependent significant differences were observed in NCD, BAI_B, BAB_B, DAN_B, MBC_B, YARW_B, KWA_B, DAN_B, BAD_B, FARO_B, and JIR_B (Appendix A).

The fruit flies on BR diets had similar catalase activity to the NCD group. In addition, all fruit flies on the WR diets had comparable catalase activity to the HFD groups in both male and female groups. Analysis of catalase activity showed that there was a significant difference (*p* < 0.05) amongst the BR, WR, and HFD groups for both male and female flies (Figure 3 and Figure 4). Moreover, sex-dependent significant differences were observed in NCD, AKM, BAI_B, BAB_B, DAN_B, MBC_B, YARW_B, YARK_B, DANB_B, BAD, FARO_B, and YARM_B.

All fruit flies on BR diets had similar MDA levels to those of the NC group. Similarly, no differences were observed between the WR groups and the HFD group. However, significant differences (*p* < 0.05) were found between the BR, WR, and HFD groups for both male and female flies (Figure 3 and Figure 4). In addition, sex-dependent significant differences were observed in AKM_W, BAI_B, DAN_W, JEEP_W, JAM_W, MBC, YARW_W, KWA_W, YARK_W, DANB_W, FARO_W, JIR_W, and YARM_B (Appendix A).

### 2.6. Effects on Gene Expression

#### 2.6.1. Expression of Insulin Receptor Substrate (IRS)

The fruit flies of the BR groups expressed comparable *IRS* mRNA relative to NCD (Figure 5 and Figure 6). Similarly, those on the WR diets expressed *IRS mRNA* comparable to the HFD groups, except for *(p* < 0.05) of BAD, FARO, and JIR in female flies, which were significantly lower than others. However, there were significant differences *(p* < 0.05) between the BR and WR groups for both male and female flies (Figure 5 and Figure 6). In addition, sex-dependent significant differences were observed in all groups (Appendix A).

#### 2.6.2. Expression of Phosphoenol Pyruvate Carboxylase (PEPCK)

The BR groups had similar *PEPCK mRNA* expression for both males and females (Figure 5 and Figure 6). However, there were significant differences *(p* < 0.05) between the BR and WR groups for both male and female flies. In contrast, the WR groups expressed comparable *PEPCK mRNA*, with the exception of *(p* < 0.05) AKM in the male group, which was higher. Meanwhile, the WR of BAI, DAN, and YARK were significantly higher *(p* < 0.05) than other WR groups in the female flies (Figure 6). In addition, sex-dependent significant differences were observed in all groups except for YARK_W (Appendix A).

#### 2.6.3. Expression of Acetyl-CoA Carboxylase (ACC)

The fruit flies on BR diets expressed higher levels of *ACC mRNA* compared with the NCD group. In contrast, those on the WR diets had similar expression levels to the HFD groups, which were higher than the expression in the BR groups (Figure 5 and Figure 6). There was no significant difference *(p* > 0.05) between WR cultivars in the male groups, with the exception (*p* < 0.05) of AKM, which was higher (Figure 5). However, there were significant differences (*p* < 0.05) between the BR, WR, and HFD groups for both male and female flies (Figure 5 and Figure 6). In addition, sex-dependent significant differences were observed in all groups except for AKM_B, JIR_B, and YARM_B (Appendix A).

## 3. Materials and Methods

### 3.1. Materials

Folin–Ciocalteu was purchased from Sisco Research Laboratories Pvt. Ltd., Mumbai, India. Distilled water was prepared using distiller machine manufactured by Laboid international, Solan, Pradesh, India. Boric acid was purchased from Central Drug House Ltd., New Delhi, India. Gallic acid and quercertin were purchased from Suvidhi Chemical Industry, Gujarat, India. Hexane, isopropanol and ascorbic acid were purchased from Changzhou Baixing Chemical Co., Ltd., Tianjin, China. Corn flour was purchased from Faso Farine Mais Blanc (FFMB), Ouagadougou, Burkina Faso. Coconut oil was purchased from Khong Guan vegetable oil refinery, Butterworth, Malaysia. Flavonoid assay kit was purchased from oxiselect™ Cell Biolabs, Inc. San Diego USA. Phosphate buffer saline was purchased from Biocept Ltd., Allschwil, Switzerland. Triglycerides, malondialdehyde (MDA), superoxide dismutase (SOD), catalase (CAT), glucose, trehalose, and glycogen assay kits were purchased from Solarbio Life Science, Beijing, China. Nucleic acid isolation kit purchased from Daan Gene Co., Ltd. of Sun Yat-sen University, Guangdong, China. Sybr-Green qRT-PCR kit (Master Mix PCR) was purchased from Toroivd Technology Company Ltd., Shanghai, China. All solvents used were of analytical grade and were purchased from Merck (Loba-chemie pvt. Ltd., Mumbai, India).

### 3.2. Sample Preparation

Fifteen commonly consumed cultivars of paddy rice in North West Nigeria were used in this study. The cultivars were *Akai maza hajj* (AKM), *Baingila* (BAI), *Baburashi* (BAB), *Jamila* (JAM), *Maibakincarki* (MBC), *Yarwasagi* (YARW), *Kwandala* (KWA), *Yarkatabore* (YARK), *Danboto* (DANB), *Bakindanboto* (BAD), *Faro 44* (FARO), *Jirkita* (JIR), *and Yarkukuma* (YARM), which were collected from Sokoto state, Nigeria, while the *Dankaushi* (DAN) and *Jeep* (JEEP) cultivars were collected from Kebbi and Kano States of Nigeria, respectively. All rice cultivars were collected from farmers at the points of harvest and transported to the laboratory in sacs for storage at ambient temperature prior to the study. The paddy rice cultivars were de-husked to obtain BR, where some were further polished into WR. Subsequently, the BR and WR were pulverised into powder, sieved, and stored in polythene bags at room temperature for further studies.

### 3.3. Proximate Analysis and Crude Fibre Content Determination of the Rice Cultivars

The proximate composition of the rice samples was determined as described previously [16,24] with modifications, following the official methods of the Association of Official Analytical Chemists [25]. Thus, contents such as ash, moisture, crude protein, fat, and crude fibre were measured, while carbohydrate content was estimated by the difference method (i.e., summation of all other percentage compositions and deducting the value from 100).

### 3.4. Analyses of the Bioactive Compound Components of the Rice Cultivars

#### 3.4.1. Preparation of WR and BR Methanolic Extracts

Fifty grams (50 g) of pulverised WR and BR were separately extracted with 500 mL of methanol by maceration for 72 h, after which the extracts were filtered using a piece of clean, sterile, white muslin cloth to remove the debris and further filtered on a Whatman filter paper No. 1. The filtrate was concentrated under reduced pressure, and the methanolic extracts obtained were tested for total phenolic and flavonoid content.

#### 3.4.2. Determination of Total Phenolic Content (TPC)

The total phenolic content of the extracts was determined using a modified Folin–Ciocalteu colorimetric method (Sisco Research Laboratories, Mumbai, India) according to the manufacturer’s instructions, with little modification. Briefly, 500 μL of freshly prepared 10% Folin–Ciocalteu reagent and different concentrations of 150 μL of the extracts or gallic acid (positive control) were incubated for 5 min. Then, the reaction was neutralised with 1 mL saturated sodium carbonate. The absorbance was measured at 760 nm after 40 min incubation, and the results were expressed as milligrams of gallic acid equivalent per 100 g of the weight of the extract (mg GAE/g DW).

#### 3.4.3. Determination of Total Flavonoid Contents (TFC)

The total flavonoid content was determined using the Dowd method as adapted by Arvouet-Grand [26] and reported by others [16,27,28]. Briefly, 2 mL of 2% aluminium trichloride (AlCl_3_) (Changzhou Baixing, Tianjin, China) in methanol was mixed with equal volume of the extract and incubated at room temperature for 10 min before reading the absorbance at 415 nm against a blank sample consisting of 2 mL of extract and 2 mL of methanol without AlCl_3_. The total flavonoid content was determined using a standard curve with quercetin (Sisco Research Laboratories, Mumbai, India) as the standard. The mean of three readings was expressed as milligrams of quercetin equivalent per gram of the weight of extract (mg QE/gDW).

#### 3.4.4. Determination of Total Oryzanol Content (TOC)

The rice sample was pulverised, sieved, and weighed before it was soaked in 50 mL of distilled water in a test tube with 2 g of ascorbic acid. The mixture was then vortexed and incubated at 40 °C for 40 min before adding 75 mL of hexane:isopropanol (1:3). The mixture was then vortexed again and centrifuged with a floor model refrigerated centrifuge machine (MX-301 Highspeed, Tomy Kogyo Co., Ltd., Tagara, Japan) at 1320× *g* for 15 min. A separator funnel was used to remove the organic layer. The residue was further re-extracted with 10 mL of hexane:isopropanol (1:3), and the entire process was repeated. The rotary evaporator (LB-R1000, Labpre Instrument and Equipment Co., Ltd., Henan, China) was set at 70 °C to evaporate the combined organic layers. The extract was dissolved in isopropanol, and its absorbance at 326 nm was recorded using a spectrophotometer (Multiskan EX, Thermo Scientific, Vantaa, Finland), and the results were expressed as mg/DW of the sample [29].

### 3.5. Fruit Fly Husbandry and Experimental Design

Wild-type (w^1118^ strain) *Drosophila melanogaster* were acquired from the Drosophila Laboratory of the Centre for Advanced Medical Research and Training (CAMRET), Usmanu Danfodiyo University, Sokoto. They were maintained at room temperature (22 to 25 °C) on a standard diet in media bottles in a natural light/dark cycle. To establish the possible effects of WR and BR consumption on metabolic parameters, young adult fruit flies of about a day old were separated into 64 groups. Each group had five replicates with 20 flies per replicate. This made a total of 100 fruit flies per group. Each group was assigned to experimental diets for 7 days as follows: normal control diet (NCD) (2 groups {males and females}), high-fat diet (HFD) (2 groups {males and females}), 15 cultivars of brown rice (30 groups (15 males and 15 females)), corresponding 15 white rice (30 groups (15 males and 15 females)) (Figure 7). The NCD is a standard cornmeal diet that contains corn flour, yeast, agar-agar, and distilled water. The HFD is a formulation composed of 90% NCD and 10% coconut oil (CO), as described by Heinrichsen [30], while rice diet formulations are composed of 50% NCD and 50% BR/WR. After seven days of the intervention, the fruit flies were assessed for negative geotaxis (climbing speed) and weight. Finally, the fruit flies were anaesthetized on ice and preserved at −20 °C until further analyses.

### 3.6. Body Weight Measurements

The weights of the fruit flies were taken before and after the intervention (7 days) using a Kern analytical weighing scale (Kern & Sohn Ltd., Balingen, Germany). Thirty (30) flies per group were anaesthetized on ice and then placed inside a pre-weighed empty microtube (Wuhan Servicebio Technology Co., Ltd., Wuhan, China) and re-weighed. The differences in weights were recorded in milligrams (mg).

### 3.7. Negative Geotaxis Assay

Thirty flies per interventional diet were anaesthetized on ice and placed in empty 50 mL measuring cylinders to assess locomotor activity [31]. The cylinder was marked at 6 cm from the bottom. Flies were allowed to acclimatise at room temperature for 10 min. The cylinder was gently tapped such that all the flies were at the bottom of the tube, and the number of flies that crossed the 6 cm mark in ten seconds was recorded. This was repeated thrice on the 7th day of exposure to interventional diets for each group to determine the average pass rate per group with 2 min resting time.

### 3.8. Biochemical Analyses

Thirty flies per interventional diet were fasted and anaesthetized on ice. They were thereafter placed in a dish and rinsed with 100 µL of cold PBS. The flies were then homogenised using 200 µL of cold PBS on ice. The homogenates were centrifuged for 3 min at 14,000× *g* in a floor model refrigerated centrifuge (MX-301 Highspeed, Tomy Kogyo Co., Ltd., Tagara, Japan) at 4 °C. The supernatant containing the haemolymph was collected for biochemical analyses.

#### 3.8.1. Glucose Assay

Fasting glucose levels of haemolymph were quantified using the Glucose Oxidase (GO) enzymatic assay kit (Spinreact, Girona, Spain) following manufacturer instructions, and the results were expressed as mg/dL.

#### 3.8.2. Trehalose Assay

The trehalose level of haemolymph was quantified using the Anthrone colorimetric kit (Solarbio Life Science, Beijing, China) in accordance with manufacturer instructions and the results were expressed as mg/g sample.

#### 3.8.3. Glycogen Assay

The glycogen level of haemolymph was also determined using the Anthrone colorimetric kit (Solarbio Life Science, Beijing, China) in accordance with manufacturer instructions, with the results expressed as mg/g sample.

#### 3.8.4. Triglyceride Assay

The triglyceride level of the haemolymph was quantified using a colorimetric kit (Spinreact, Girona, Spain) according to the manufacturer’s instructions. 

#### 3.8.5. Anti-Oxidant Markers Assay

Malondialdehyde (MDA) level and activities of superoxide dismutase (SOD) and catalase (CAT) in the haemolymph were determined using colorimetric assay kits (Solarbio Life Science, Beijing, China) according to the manufacturer’s instructions.

### 3.9. Gene Expression Analysis

#### 3.9.1. Extraction of RNA

RNA was extracted from 10 flies per group, using the nucleic acid isolation kit (Daan Gene Co., Ltd., Sun Yat-sen University, Guangdong, China) according to the manufacturer’s instructions. Briefly, lysis buffer was added to the microtube containing the flies, and the flies were homogenised using a pellet pestle. Thereafter, chloroform was added to the homogenate, which was then centrifuged in a floor model refrigerated centrifuge machine (MX-301 Highspeed, Tomy Kogyo Co., Ltd., Tagara, Japan) at 12,000× *g* at 4 °C for 1 min. The upper aqueous layer was then transferred into a new microtube, followed by the addition of absolute ethanol. The mixture was then transferred to the RNA binding column and subjected to centrifugation at 12,000× *g* at 4 °C for 1 min. After discarding the flow-through, wash buffer 1 was added and spun at 12,000× *g* at 4 °C for 1 min. The flow-through was discarded, and another wash buffer 2 was added and spun again at 12,000× *g* at 4 °C for 1 min. After discarding the flow-through, the binding column was spun dry at 12,000× *g* at 4 °C for 3 min and transferred to a new microtube. Approximately 50 µL of eluent buffer was added and soaked for 3 min and then spun at 12,000× *g* at 4 °C for 1 min. The extracted RNA was considered acceptable when it had purity readings for A260/230 and A260/280 between 1.8 and 2.0 using a BioSpec-nano spectrophotometer (Shimadzu Biotech, Japan).

#### 3.9.2. Primer Design

Primer quest tool software was used to design primers using *Drosophila melanogaster* gene sequences obtained from the National Centre for Biotechnology Information Gen Bank Database [32]. The genes of interest were *dPEPCK*, *dIRS*, and *dACC*, and *RPL-32* was used as the housekeeping gene (Table 4). 

#### 3.9.3. Quantitative Real-Time Polymerase Chain Reaction (RT-qPCR) Analysis

RT-qPCR of RNA samples was carried using a one-step SYBR Green Master Mix (Toroivd Technology Company Ltd., Shanghai, China) according to the manufacturer’s protocol. Briefly, the reaction mixture was prepared by mixing RNA samples (2 µL each), 10 µL of qRT PCR master mix, 1 µL of manganese (Mn), 0.4 µL of forward/reverse primers, and 6.2 µL DNAse/RNase-free water (PCR grade water). The mixture was gently vortexed and loaded on a Rotor-Gene Q-5plex platform thermal cycler (Qiagen, Hilden, Germany) set with the following cycling conditions: denaturation at 90 °C for 30 s, reverse transcription at 61 °C for 20 min, pre-denaturation at 95 °C for 1 min, followed by 45 cycles involving denaturation at 95 °C for 15 s, annealing at 53 °C for 15 s and extension at 74 °C for 18 s. The fold change for each gene was determined using the 2^−ΔΔCT^ method [33].

### 3.10. Statistical Analysis

All data were subjected to Shapiro–Wilk normality test to determine the choice of statistical analyses; parametric tests were used to analyse results. The results were expressed as means ± standard deviations (SD). A one-way analysis of variance (ANOVA) followed by Bonferroni post hoc test were used to determine the significant differences in proximate analysis and bioactive compound components. Two-way ANOVA and *Bonferroni’s* multiple comparisons test were used to determine the sex-dependent differences in body weight, negative geotaxis, biochemical analysis, and gene expression analysis. Differences were considered statistically significant at *p* < 0.05. All analyses were performed using IBM SPSS Statistics (Version 20.0. IBM Corp., Armonk, NY, USA).

## 4. Discussion

Currently, local rice cultivars are the major staple food, providing energy and nutrients for the large Nigerian population. The Nigerian locally grown BR cultivars used in this study have an abundant protein composition ranging from 10.90 ± 0.1% to 11.67 ± 0.10%, which is higher than previously reported values by Oko and Ugwu [34] (1.58 ± 0.01% to 6.22 ± 0.01%) and Odenigbo [35] (8.26 ± 0.20% to 10.52 ± 0.35%). In contrast, the WR cultivars have low protein contents, which vary from 2.43 ± 0.20% to 10.4 ± 0.20% as a result of the milling process.

Furthermore, the BR samples in our study have high quantities of carbohydrates, ranging from 64.66 ± 0.67% to 70.22 ± 0.64%. These values are lower than the values obtained by Odenigbo [35] (87.79% to 89.44%) and Oko and Ugwu [34] (76.92% to 85.09%), both of whom analysed the proximate compositions of BR in South East Nigeria. The low carbohydrate content may be attributed to environmental factors, such as climate and planting methods [36,37]. The carbohydrate content of the WR was similar to those of the BR, likely because polishing has no effect on the carbohydrate composition of rice. Carbohydrates are the main determinant of the glycaemic index of rice, an indicator of the amount of glucose available for energy production for immediate cellular metabolism or storage for subsequent utilization when needed. Accordingly, consumption of WR as a staple diet causes sustained hyperglycaemia, which could deteriorate into metabolic diseases, such as obesity and IR [14].

The percentage of fibre content among the fifteen BR samples was higher than the range (1.5 to 2.0%) reported by Oko and Ugwu [34] in a study conducted in Ebonyi, South East Nigeria. The polishing of BR generally decreases its fibre content. The fibre contents of WR samples in our study were lower than those of BR. Dietary fibre can slow down the digestion of starches and the absorption of glucose into the bloodstream, thus preventing the postprandial hyperglycaemia that accompanies the consumption of high GI diets such as WR and reducing the risks of metabolic diseases [12,38]. The ash contents of the rice samples are indicative of the inorganic elements present in the samples. The higher ash contents in the WR suggest the presence of higher levels of nutritionally non-essential contents in comparison to the BR. Odenigbo [35] reported a range of 0.23 ± 0.32% to 0.80 ± 0.12% in their study conducted on BR rice varieties from Abakaliki, South East Nigeria, which is different from the results obtained in this study. Therefore, consuming BR of these cultivars along with other healthy diets makes it easier to achieve the recommended daily allowance/adequate intakes, which is estimated for individuals aged 19–50 years at 56 g/day (male) and 46 g/day (female) for protein, 130 g/day for carbohydrate, and 38 g/day (male) and 25 g/day (female) for dietary fibre [39].

The percentage moisture contents for both BR and WR cultivars ranged from 7.33 ± 0.77% to 9.5 ± 0.5%, which is different from what Oko and Ugwu [34] reported (3.67 ± 0.01 to 18.0 ± 0.1). Moisture content may be attributed to the degree of temperature the rice was exposed to previously, and this affects the milling, storage, and taste of rice and is a major determinant of rice shelf life [34]. The percentage lipid content of rice cultivars in this study ranged from 0.17 ± 0.29 to 2.83 ± 0.29, which is consistent with values reported by Oko and Ugwu [34] (0.5% to 3.5%) but not by Wordu and Banigo [38] (0.3% to 1.6%). The disparity in our proximate composition with other studies could be attributed to differences in factors such as location, paddy soil, planting methods, climate, harvest time, handling, and storage of rice samples before the analysis methods [35].

Oryzanols have been demonstrated to have anti-cholesterol effects mediated via reducing hepatic fat accumulation and inflammation [7,10]. More so, it has anti-glycaemic effects mediated via enhancing glucose uptake and GLUT-4 translocation to the cell surface [40,41,42,43,44]. Its antioxidant and anti-diabetic effects have equally been reported, which are mediated by inhibiting rapid digestion and absorption of glucose in the intestine as well as by preventing the accumulation of ROS in the tissues [42,43,45,46]. The BR samples in our study had high quantities of oryzanol contents, consistent with the average value (45 mg/g) reported for local BR from Chiang Mai, Thailand, by Hongsibong et al. [43]. As a result of the polishing processes, WR had low oryzanol contents compared with the corresponding BR. Furthermore, the BR samples used in this study are rich in polyphenols, such as flavonoids and other phenolic compounds. The flavonoid content observed in this study was similar to that reported by Salawu et al. [47] (2.9 g% to 11.93 g%) in a study conducted on rice cultivars from the North Central and South West regions of Nigeria. However, the results are different from those reported by Oselebe [5] (0.5 to 3.0 g%). 

The flavonoid contents of WR were lower than those of BR due to the loss of the bran layer to the polishing process. It is well known that flavonoids possess antioxidant, anti-inflammatory, and hepatoprotective effects [48]. They also cause inhibition of glucose absorption, thereby producing lower glucose levels and inhibiting gluconeogenesis and glycogenolysis [5,45,48,49]. The total phenolic contents of the fifteen cultivars of BR were similar to those reported by Oselebe [5] (0.09 to 0.75 mg GAE/g DW). The lower values observed for WR may have been due to the loss of the bran layer. Moreover, the bran layer of BR is rich in phenolics that confer anti-inflammatory and antioxidant effects. In addition, phenolic acid inhibits the rapid absorption of glucose, thereby decreasing postprandial hyperglycaemia, enhancing insulin release, activating insulin receptors, and modulating glucose release from the liver. All of these can protect against diet-induced metabolic disorders [44,47,49,50,51]. Studies have linked diet-induced metabolic disorders such as obesity to WR consumption [8,52,53,54,55]. In this study, fruit fly consumption of WR appeared to promote fat expansion, weight gain, higher oxidised LDL levels, and elevated oxidative stress that were observed. Similar observations were also made by [11], who reported enhanced body weight gain in rats fed on WR. The reduced body weight seen in the BR groups can be attributed to the multiple bioactive compounds in the BR cultivars. Ho et al. and Bui et al. [56,57] also reported that BR suppressed the weight and size of epididymal adipose tissue and significantly decreased lipid accumulation in the liver. This is supported by evidence from human studies where BR reduced body weight and increased HDL levels [58].

A negative geotaxis assay is an assessment of the fly’s ability to move against gravity. This has been used as an indicator of obesity, Alzheimer’s, and Parkinson’s diseases in *Drosophila* models [59,60]. In this study, the decreased locomotor activity, as measured by the negative geotaxis assay, observed in the WR and HFD groups, is likely due to an increase in body weight. This is consistent with the study by Trindade et al. [61] who reported a decrease in the negative geotaxis ability of flies following 7 days of exposure to HFD. The BR groups had good locomotor activity, suggesting that the bioactive compounds in BR may have contributed to this property.

In *Drosophila*, the dominant haemolymph sugar is the disaccharide trehalose, which is synthesised by trehalose-6-phosphate synthase 1 in the fat body (analogous to the mammalian liver) [62]. Simple sugars from consumed food are taken up passively from the digestive tract and directly into the fat body, where they are converted to trehalose, a non-reducing sugar, and where they are stored or released into the haemolymph as the primary circulating sugar [23]. In this study, high glucose and trehalose concentrations were observed in the WR and HFD groups. This is consistent with the study conducted by Musselman et al. [63] who demonstrated that chronic consumption of high-sugar diets elevated haemolymph glucose and trehalose concentrations compared with other high-calorie diets. However, in comparison to WR, the availability of abundant bioactive compounds in BR may have helped to maintain glucose homeostasis in fruit flies, as was evident in our study where the BR groups had relatively similar concentrations.

Studies have linked insulin resistance (IR) to WR consumption [8,52,53,54,55]. IR is a metabolic disorder characterised by chronic hyperglycaemia caused by abnormal glucose use in skeletal muscle and increased hepatic glucose production via glycogenolysis or gluconeogenesis [64]. This occurs due to the inhibition of insulin-stimulated glucose uptake, which causes tissue starvation, thereby inhibiting glycogen synthesis and enhancing the conversion of stored glycogen back to glucose, which eventually worsens the hyperglycaemic condition [65]. Excess energy is stored in the form of glycogen in the skeletal muscle and the liver during a normal nutritional state for utilisation during deficiency when it would be mobilised and released via glycogenolysis [66]. In this study, a decrease in glycogen levels was observed in the WR groups, in addition to the elevated fasting glucose reported in the same groups earlier. This is indicative of ongoing glycogenolysis in response to tissue energy starvation, which occurs in IR. This is consistent with other studies, which reported a decrease in glycogen synthesis from the downregulation of glucokinase (GCK) as a result of WR consumption [9,64,65]. However, higher glycogen levels were observed in the BR groups due to active glycogenesis; this is supported by the study conducted by Gao et al. [67] where GCK was upregulated in the BR group and downregulated in the WR group.

Impairment in the insulin signalling pathway results in a deficiency of insulin stimulation of glucose uptake leading to IR [68]. Existing evidence suggests that serine/threonine phosphorylation attenuates insulin signalling by reducing the extent of tyrosine phosphorylation of *IRS* proteins [69,70]. Our study demonstrated a decline in *IRS mRNA* expression in the WR groups, which clearly showed impairment in the insulin signalling pathway, and partly explains the elevated glucose and trehalose concentrations. However, increased *IRS mRNA* expression was observed in the BR groups, suggesting an adequate flow of insulin signalling pathways. This is consistent with the studies by Shen et al. [64] and Adamu et al. [68], both of whom reported upregulation of *IRS mRNA* expression in the BR groups of the rat model.

As a result of the inhibition of insulin-stimulated glucose uptake, tissues resort to stored carbohydrates by activating glycogenolysis for immediate compensation, or non-glucose compounds by activating gluconeogenesis for long-term compensation [71,72]. The initial gluconeogenic step is catalysed by *PEPCK*, which is upregulated in the WR groups in this study. This is consistent with our previous report on the upregulation of *PEPCK mRNA* expression by WR and its downregulation by BR [71].

Sex-dependent differences were observed in the glucose metabolism of some WR groups as evident in trehalose concentration and glycogen accumulation; this was supported by the downregulation of *IRS mRNA* and by the upregulation of *PEPCK mRNA*. In addition, male androgen hormones were implicated due to their ability to suppress leptin secretion, which increased the rate of consumption of WR and led to persistent hyperglycaemia [73]. However, oestrogen was reported to stimulate the secretion of leptin, which reduced the consumption rate of female flies [74].

Accordingly, the current study demonstrates how the consumption of Nigerian locally grown BR and corresponding WR cultivars affect carbohydrate metabolism in fruit flies. Regular consumption of WR promotes persistent low-grade inflammation, which enhances serine/threonine phosphorylation [71,75,76,77] and downregulation of the *mRNA* expression of *IRS*, thereby inhibiting insulin-stimulated glucose uptake and causing IR. Sustained IR enhances the breaking down of stored glycogen in the liver via glycogenolysis as well as via upregulation of the *mRNA* expression of *PEPCK*, which enhances the synthesis of glucose from lipids and proteins to meet tissue glucose demands. Both gluconeogenesis and glycogenolysis replenish the tissues with energy to prevent starvation. However, the processes rather worsen the hyperglycaemic conditions, which if not properly managed, advances into T2DM.

Most obese individuals have elevated plasma levels of FFA and TG which are known to cause peripheral IR [10,78]. In this study, elevated triglyceride levels were observed in WR and HFD groups, in agreement with the findings of Trindade et al. [61], who demonstrated elevated triglyceride levels in flies exposed to HFD. Meanwhile, BR produced lower triglyceride levels likely due to the presence of bioactive compounds, such as oryzanol, flavonoid and phenolics in the BR cultivars which possess anti-cholesterol and anti-obesity effects. This is consistent with our previous study [10] where BR was demonstrated to lower total cholesterol, triglyceride and LDL-C in a rat model.

The Drosophila Acetyl-coA Carboxylase (*dACC*) gene has been identified as the homologue of the mammalian *ACC1* or *ACC2* coding sequences [79]. It encodes the rate-limiting enzyme, which catalyses the carboxylation of acetyl-CoA to malonyl-CoA in the synthesis of fatty acid. Increased levels of circulatory fatty acid and their accumulation as triglycerides (TGs) in the fat body constitute a critical step in the development of obesity and type 2 diabetes [79,80]. Sustained postprandial hyperglycaemia enhanced the conversion of excess glucose to FFA and TG via lipogenesis by activation and upregulation of *dACC mRNA* [81]. This is evident in this study where WR and HFD groups had upregulation of *dACC mRNA*, which explains the reason for the high TG concentration. However, downregulation of *dACC mRNA* was observed in BR groups in comparison to the NC groups, which is an indication of reduced accumulation of lipids, as demonstrated by the lower TG levels. This is consistent with the study by Ho et al. [13] who reported downregulation of *ACC mRNA* and reduced fat accumulation in BR and germinated BR groups compared to the WR group.

Furthermore, a decrease in SOD and CAT activities, and an increase in MDA in both WR and HFD groups in this study is consistent with the study by Wunjuntuk et al. [82], who reported a similar trend of activities in the WR group compared to the BR group. Meanwhile, obesity and OS are positively associated with each other, and the elevation of ROS synthesis enhances fat accumulation [80]. Regular consumption of WR has been demonstrated to facilitate ROS production as a result of enhanced production of FFA and TG [10], whereas BR improves antioxidant status [83]

Moreover, sex-dependent differences were observed in the WR groups of the male category, showing higher percentage of TG gain, weight changes, and *dACC mRNA* compared with the corresponding WR groups in the female category. This could be explained by the ability of male androgen hormones to suppress the secretion of leptin (hunger inhibitor). This, therefore, increased the rate of consumption of WR, leading to persistent hyperglycaemia, most of which underwent lipogenesis as a result of upregulation of ACC mRNA expression, with consequent increases in TG, FFA, fat accumulation, and weight gain among the male flies in the WR groups [73]. However, it has been reported that oestrogen stimulates leptin hormone secretion, which reduces the consumption rate of female flies and consequently less TG, FFA, and weight gain [74].

Previous studies have shown that each of the bioactive compounds has a positive effect on metabolic health [45,48]. Thus, the synergistic effects of γ-oryzanol, flavonoids, and phenolics may have played a role in the outcomes observed in the current study. Individual bioactive compounds are usually arranged in a highly complex manner that may increase or decrease their effects in the body [84,85], resulting in a food synergy effect.

## 5. Conclusions

Findings from this study have shown that the selected Nigerian WR cultivars could promote metabolic perturbations via hyperglycaemia, lipogenesis, adiposity, and excessive weight gain in fruit flies. However, BR cultivars were shown to have anti-glycaemic, anti-lipogenesis, and anti-adiposity properties, which could be attributable to the availability of abundant bioactive contents in their bran. Mechanistically, the probable synergistic effects of the bioactive compounds in BR were seen in suppressed gluconeogenesis (*dPEPCK*) and glycogenolysis and enhanced insulin signalling pathways as indicated by *dIRS* upregulation. From the foregoing, the composition of BR, if present in a rice cultivar, as demonstrated in the BR of *Akai maza hajji*, *Baburashi*, *Ba’ingila*, *Dan kaushi*, and *Jeep* cultivars, would prevent individuals who depend on them as a staple food from diet-related metabolic diseases.

## Figures and Tables

**Figure 1 molecules-28-00532-f001:**
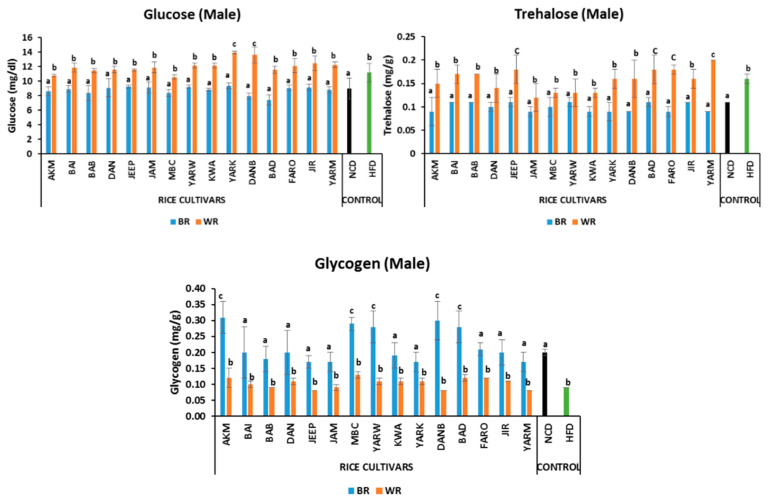
Effects of rice cultivars on the levels of fasting glucose, trehalose and glycogen in male flies after 7 days of intervention diets. Bars represent mean ± SD. Bars denoted by the same letters are not significantly different (*p* > 0.05). (n = 30). Black and green bars represent the groups fed normal diet and high fat diet respectively. NCD = normal diet, HFD = high-fat diet *Akai maza hajj* = AKM, *Baingila* = BAI, *Baburashi* = BAB, *Jamila* = JAM, *Maibakincarki* = MBC, *Yarwasagi* = YARW, *Kwandala* = KWA, *Yarkatabore* = YARK, *Danboto* = DANB, *Bakindanboto* = BAD, *Faro 44* = FARO, *Jirkita* = JIR, *Yarkukuma* = YARM, *Dankaushi* = DAN, *Jeep* = JEEP.

**Figure 2 molecules-28-00532-f002:**
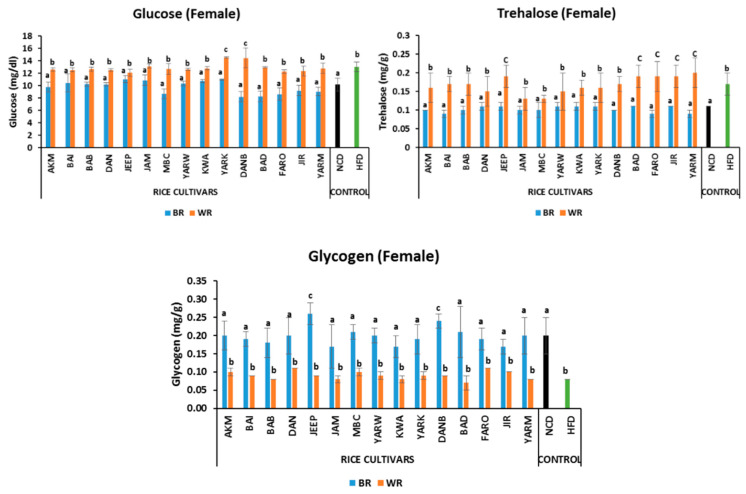
Effects of rice cultivars on the levels of fasting glucose, trehalose and glycogen in female flies after 7 days of intervention. Bars represent Mean ± SD. Bars denoted by the same letters are not significantly different (*p* > 0.05). (n = 30). Black and green bars represent the groups fed normal diet and high-fat diet respectively. NCD= normal diet, HFD = high-fat diet *Akai maza hajj* = AKM, *Baingila* = BAI, *Baburashi* = BAB, *Jamila* = JAM, *Maibakincarki* = MBC, *Yarwasagi* = YARW, *Kwandala* = KWA, *Yarkatabore* = YARK, *Danboto* = DANB, *Bakindanboto* = BAD, *Faro 44* = FARO, *Jirkita* = JIR, *Yarkukuma* = YARM, *Dankaushi* = DAN, *Jeep* = JEEP.

**Figure 3 molecules-28-00532-f003:**
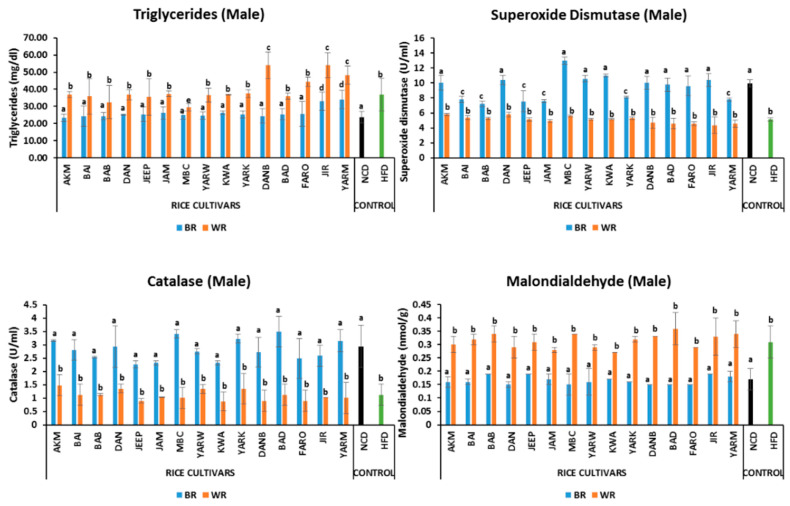
Effects of rice cultivars on triglyceride level and oxidative stress markers in male flies after 7 days of intervention. Bars represent mean ± SD. Bars denoted by the same letters are not significantly different (*p* > 0.05). (n = 30). Black and green bars represent the groups fed normal diet and high fat-diet respectively. NCD = normal diet, HFD = high-fat diet, *Akai maza hajj* = AKM, *Baingila* = BAI, *Baburashi* = BAB, *Jamila* = JAM, *Maibakincarki* = MBC, *Yarwasagi* = YARW, *Kwandala* = KWA, *Yarkatabore* = YARK, *Danboto* = DANB, *Bakindanboto* = BAD, *Faro 44* = FARO, *Jirkita* = JIR, *Yarkukuma* = YARM, *Dankaushi* = DAN, *Jeep* = JEEP.

**Figure 4 molecules-28-00532-f004:**
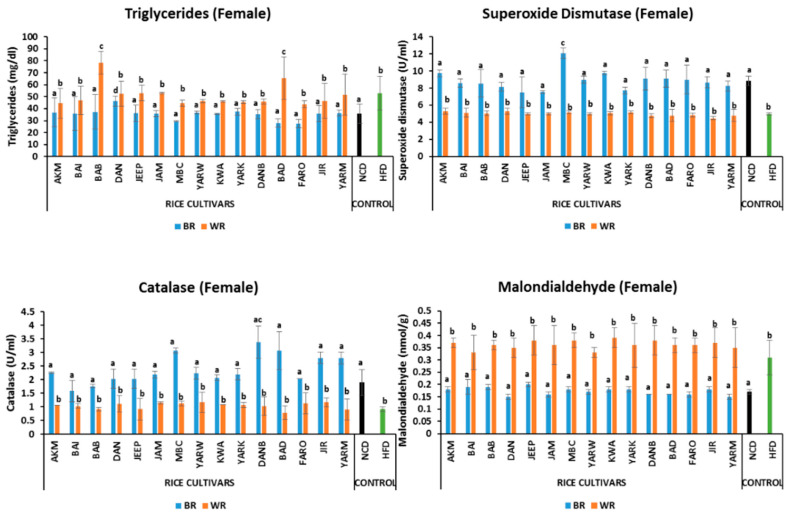
Effects of rice cultivars on triglyceride level and oxidative stress markers in female flies after 7 days of intervention. Bars represent mean ± SD. Bars denoted by the same letters are not significantly different (*p* > 0.05). (n = 30). Black and green bars represent the groups fed normal diet and high-fat diet respectively. NCD = normal diet, HFD = high-fat diet, *Akai maza hajj* = AKM, *Baingila* = BAI, *Baburashi* = BAB, *Jamila* = JAM, *Maibakincarki* = MBC, *Yarwasagi* = YARW, *Kwandala* = KWA, *Yarkatabore* = YARK, *Danboto* = DANB, *Bakindanboto* = BAD, *Faro 44* = FARO, *Jirkita* = JIR, *Yarkukuma* = YARM, *Dankaushi* = DAN, *Jeep* = JEEP.

**Figure 5 molecules-28-00532-f005:**
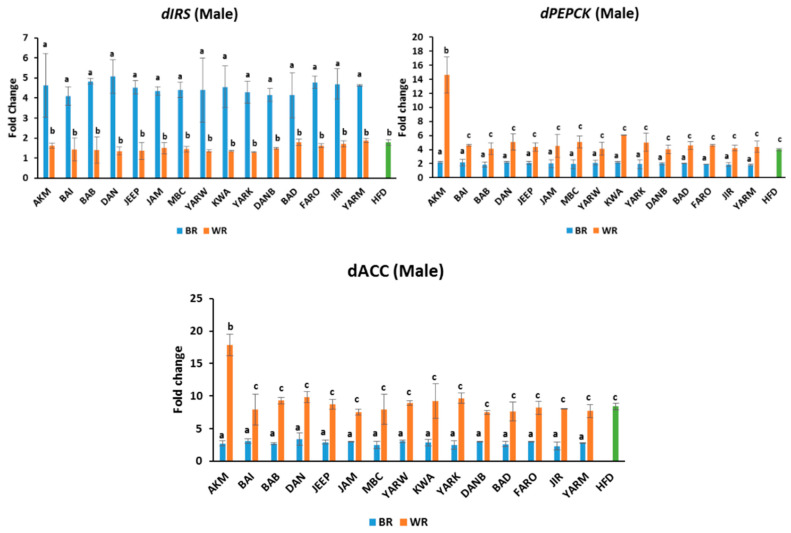
*dIRS*, *dPEPCK*, and *dACC* expressions in male flies exposed to different rice cultivars for 7 days. Fold change was calculated using Livak method (2^−ΔΔCT^). Bars represent mean ± SD. Bars denoted by the same letters are not significantly different (*p* > 0.05). (n = 20). Green bar represent the group fed high-fat diet. HFD = high-fat diet, *Akai maza hajj* = AKM, *Baingila* = BAI, *Baburashi* = BAB, *Jamila* = JAM, *Maibakincarki* = MBC, *Yarwasagi* = YARW, *Kwandala* = KWA, *Yarkatabore* = YARK, *Danboto* = DANB, *Bakindanboto* = BAD, *Faro 44* = FARO, *Jirkita* = JIR, *Yarkukuma* = YARM, *Dankaushi* = DAN, *Jeep* = JEEP.

**Figure 6 molecules-28-00532-f006:**
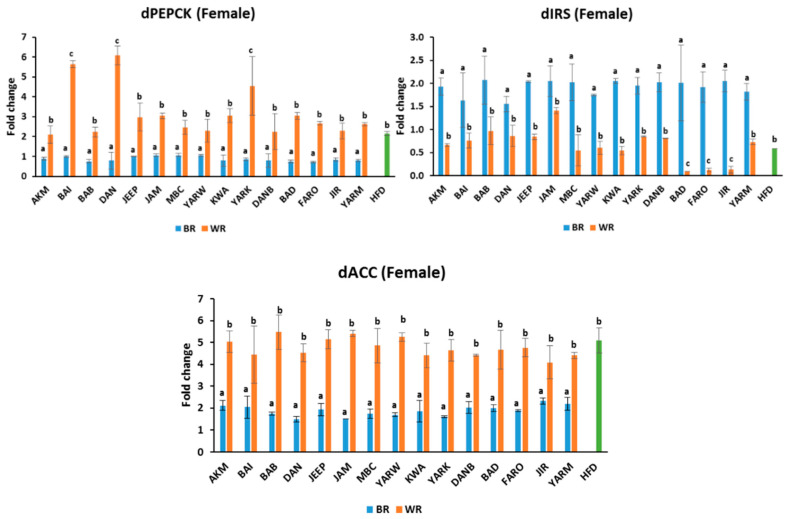
*dIRS*, *dPEPCK*, and *dACC* expressions in female flies exposed to different rice cultivars for 7 days. Fold change was calculated using Livak method (2^−ΔΔCT^). Bars represent mean ± SD. Bars denoted by the same letters are not significantly different (*p* > 0.05). (n = 20). Green bar represent the group fed high-fat diet. HFD = high-fat diet, *Akai maza hajj* = AKM, *Baingila* = BAI, *Baburashi* = BAB, *Jamila* = JAM, *Maibakincarki* = MBC, *Yarwasagi* = YARW, *Kwandala* = KWA, *Yarkatabore* = YARK, *Danboto* = DANB, *Bakindanboto* = BAD, *Faro 44* = FARO, *Jirkita* = JIR, *Yarkukuma* = YARM, *Dankaushi* = DAN, *Jeep* = JEEP.

**Figure 7 molecules-28-00532-f007:**
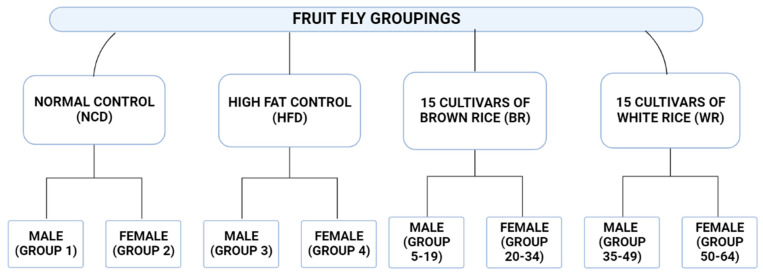
Fruit fly groups.

**Table 1 molecules-28-00532-t001:** Proximate composition and crude fibre content of rice cultivars in dry weight.

S/N	RICE CULTIVARS	% PROTEIN	% CARBOHYDRATE	% MOISTURE	% ASH	% LIPID	% FIBRE
**1**	AKM_B	11.26 ± 0.10 ^a^	70.07 ± 1.07 ^a^	7.83 ± 0.30 ^a^	4.83 ± 0.28 ^a^	2.67 ± 0.30 ^a^	5.33 ± 0.77 ^a^
**2**	AKM_W	6.91 ± 0.10 ^b^	67.42 ± 0.75 ^b^	7.67 ± 0.29 ^a^	4.67 ± 0.29 ^a^	2.33 ± 0.29 ^a^	1.00 ± 0.02 ^b^
**3**	BAI_B	11.67 ± 0.10 ^a^	64.66 ± 0.67 ^c^	8.40 ± 0.87 ^a^	7.83 ± 0.30 ^b^	2.17 ± 0.05 ^a^	5.17 ± 0.25 ^a^
**4**	BAI_W	4.49 ± 0.27 ^c^	62.34 ± 0.58 ^c^	7.33 ± 0.77 ^a^	9.50 ± 0.50 ^c^	2.17 ± 0.33 ^a^	1.17 ± 0.09 ^b^
**5**	BAB_B	11.09 ± 0.09 ^a^	65.39 ± 1.79 ^c^	8.17 ± 0.28 ^a^	6.67 ± 0.58 ^b^	2.33 ± 0.27 ^a^	4.67 ± 0.30 ^a^
**6**	BAB_W	3.44 ± 0.27 ^c^	64.08 ± 0.19 ^c^	8.17 ± 0.58 ^a^	8.00 ± 0.54 ^b^	1.83 ± 0.05 ^a^	0.17 ± 0.02 ^b^
**7**	DAN_B	11.14 ± 0.09 ^a^	68.19 ± 0.54 ^b^	8.33 ± 0.29 ^a^	4.67 ± 0.27 ^a^	2.83 ± 0.25 ^a^	5.17 ± 0.58 ^a^
**8**	DAN_W	5.32 ± 0.20 ^c^	67.02 ± 0.75 ^b^	8.00 ± 0.50 ^a^	5.00 ± 0.55 ^a^	2.33 ± 0.28 ^a^	1.83 ± 0.03 ^b^
**9**	JEEP_B	11.26 ± 0.10 ^a^	66.57 ± 0.75 ^c^	8.23 ± 0.58 ^a^	7.17 ± 0.26 ^b^	2.33 ± 0.34 ^a^	4.83 ± 0.28 ^a^
**10**	JEEP_W	4.09 ± 0.09 ^c^	63.91 ± 0.42 ^c^	8.01 ± 0.05 ^a^	7.33 ± 0.32 ^b^	2.00 ± 0.29 ^a^	0.33 ± 0.08 ^b^
**11**	JAM_B	11.00 ± 0.10 ^a^	70.16 ± 0.74 ^a^	7.52 ± 0.02 ^a^	7.60 ± 0.04 ^b^	2.33 ± 0.31 ^a^	4.12 ± 0.04 ^a^
**12**	JAM_W	7.43 ± 0.11 ^b^	67.19 ± 0.02 ^b^	8.34 ± 0.07 ^a^	7.33 ± 0.56 ^b^	2.50 ± 0.01 ^a^	1.39 ± 0.27 ^b^
**13**	MBC_B	11.14 ± 0.10 ^a^	70.44 ± 0.06 ^a^	8.37 ± 0.28 ^a^	4.67 ± 0.25 ^a^	2.83 ± 0.26 ^a^	4.17 ± 0.19 ^a^
**14**	MBC_W	7.38 ± 0.18 ^b^	68.15 ± 0.03 ^b^	8.42 ± 0.30 ^a^	5.33 ± 0.57 ^a^	2.67 ± 0.31 ^a^	2.29 ± 0.20 ^b^
**15**	YARW_B	10.91 ± 0.10 ^a^	70.68 ± 0.59 ^a^	9.50 ± 0.51 ^b^	4.50 ± 0.52 ^a^	2.83 ± 0.35 ^a^	4.54 ± 0.11 ^a^
**16**	YARW_W	7.16 ± 0.10 ^b^	67.45 ± 0.57 ^b^	9.83 ± 0.57 ^b^	5.00 ± 0.45 ^a^	2.67 ± 0.27 ^a^	1.10 ± 0.00 ^b^
**17**	KWA_B	11.03 ± 0.18 ^a^	70.99 ± 0.00 ^a^	9.63 ± 0.58 ^b^	4.17 ± 0.32 ^a^	2.83 ± 0.19 ^a^	4.17 ± 0.23 ^a^
**18**	KWA_W	8.01 ± 0.10 ^b^	66.59 ± 0.43 ^b^	9.50 ± 0.55 ^b^	4.17 ± 0.30 ^a^	2.67 ± 0.24 ^a^	1.78 ± 0.02 ^b^
**19**	YARK_B	11.14 ± 0.10 ^a^	71.08 ± 0.05 ^a^	9.50 ± 0.50 ^b^	4.67 ± 0.29 ^a^	2.41 ± 0.34 ^a^	4.80 ± 0.00 ^a^
**20**	YARK_W	10.30 ± 0.47 ^b^	66.53 ± 0.35 ^b^	10.00 ± 0.52 ^b^	4.17 ± 0.26 ^a^	2.33 ± 0.36 ^a^	1.20 ± 0.01 ^b^
**21**	DANB_B	11.73 ± 0.18 ^a^	69.61 ± 0.73 ^a^	8.00 ± 0.51 ^a^	4.17 ± 0.59 ^a^	2.33 ± 0.28 ^a^	5.17 ± 0.25 ^a^
**22**	DANB_W	2.61 ± 0.10 ^c^	67.56 ± 0.67 ^b^	8.33 ± 0.28 ^a^	3.83 ± 0.48 ^a^	0.83 ± 0.30 ^b^	0.83 ± 0.07 ^b^
**23**	BAD_B	11.49 ± 0.20 ^a^	67.20 ± 1.07 ^b^	8.17 ± 0.29 ^a^	4.17 ± 0.32 ^a^	2.67 ± 0.26 ^a^	5.33 ± 0.23 ^a^
**24**	BAD_W	3.09 ± 0.09 ^c^	66.51 ± 0.09 ^b^	8.17 ± 0.58 ^a^	4.17 ± 0.28 ^a^	1.17 ± 0.04 ^b^	0.89 ± 0.01 ^b^
**25**	FARO_B	11.38 ± 0.30 ^a^	70.92 ± 0.64 ^a^	8.00 ± 0.49 ^a^	4.33 ± 0.30 ^a^	2.33 ± 0.33 ^a^	4.50 ± 0.50 ^a^
**26**	FARO_W	3.97 ± 0.10 ^c^	67.53 ± 1.01 ^b^	7.83 ± 0.28 ^a^	4.33 ± 0.33 ^a^	1.17 ± 0.02 ^b^	0.24 ± 0.08 ^b^
**27**	JIR_B	11.38 ± 0.18 ^a^	70.29 ± 0.43 ^a^	8.01 ± 0.53 ^a^	7.33 ± 0.26 ^b^	2.33 ± 0.54 ^a^	4.66 ± 0.77 ^a^
**28**	JIR_W	4.97 ± 0.10 ^c^	68.09 ± 0.86 ^b^	7.83 ± 0.28 ^a^	7.00 ± 0.50 ^b^	0.70 ± 0.03 ^b^	0.53 ± 0.09 ^b^
**29**	YARM_B	11.49 ± 0.27 ^a^	70.01 ± 0.27 ^a^	8.33 ± 0.30 ^a^	5.17 ± 0.25 ^a^	2.67 ± 0.25 ^a^	4.33 ± 0.29 ^a^
**30**	YARM_W	2.43 ± 0.20 ^c^	68.40 ± 0.93 ^b^	8.13 ± 0.27 ^a^	5.83 ± 0.57 ^a^	1.33 ± 0.09 ^b^	0.82 ± 0.08 ^b^

Mean ± SD values within a column denoted by the same letters in superscript are not significantly different (*p* > 0.05). _B = brown rice, _W = white rice (n = 3). *Akai maza hajj* = AKM, *Baingila* = BAI, *Baburashi* = BAB, *Jamila* = JAM, *Maibakincarki* = MBC, *Yarwasagi* = YARW, *Kwandala* = KWA, *Yarkatabore* = YARK, *Danboto* = DANB, *Bakindanboto* = BAD, *Faro 44* = FARO, *Jirkita* = JIR, *Yarkukuma* = YARM, *Dankaushi* = DAN, *Jeep* = JEEP.

**Table 2 molecules-28-00532-t002:** Bioactive composition of rice cultivars.

S/N	CULTIVARS	Total Oryzanol Content (mg/DW)	Total Phenolic Content (mg GAE/g DW)	Total Flavonoid Content (mg QE/gDW)
**1**	AKM_B	46.92 ± 0.12 ^a^	11.34 ± 0.02 ^a^	0.47 ± 0.03 ^a^
**2**	AKM_W	21.23 ± 0.09 ^b^	1.48 ± 0.01 ^b^	0.15 ± 0.02 ^c^
**3**	BAI_B	46.59 ± 0.57 ^a^	11.30 ± 0.21 ^a^	0.48 ± 0.01 ^a^
**4**	BAI_W	20.13 ± 0.05 ^b^	1.18 ± 0.05 ^b^	0.22 ± 0.01 ^b^
**5**	BAB_B	46.62 ± 0.13 ^a^	10.50. ± 0.01 ^a^	0.51 ± 0.04 ^a^
**6**	BAB_W	18.97 ± 0.38 ^b^	1.45 ± 0.31 ^b^	0.15 ± 0.02 ^c^
**7**	DAN_B	46.25 ± 0.08 ^a^	11.43 ± 0.04 ^a^	0.49 ± 0.07 ^a^
**8**	DAN_W	14.94 ± 0.54 ^c^	1.81 ± 0.01 ^b^	0.19 ± 0.02 ^b^
**9**	JEEP_B	47.39 ± 0.05 ^a^	11.21 ± 0.11 ^a^	0.48 ± 0.04 ^a^
**10**	JEEP_W	16.21 ± 0.12 ^c^	3.25 ± 0.02 ^c^	0.20 ± 0.00 ^b^
**11**	JAM_B	47.30 ± 0.04 ^a^	10.94 ± 0.05 ^a^	0.51 ± 0.08 ^a^
**12**	JAM_W	20.53 ± 0.08 ^b^	1.35 ± 0.08 ^b^	0.14 ± 0.01 ^c^
**13**	MBC_B	46.98 ± 0.04 ^a^	11.07 ± 0.53 ^a^	0.48 ± 0.06 ^a^
**14**	MBC_W	21.13 ± 0.55 ^b^	1.62 ± 0.01 ^b^	0.19 ± 0.04 ^b^
**15**	YARW_B	46.71 ± 0.04 ^a^	10.94 ± 0.31 ^a^	0.47 ± 0.07 ^a^
**16**	YARW_W	20.76 ± 0.08 ^b^	2.94 ± 0.01 ^c^	0.14 ± 0.01 ^c^
**17**	KWA_B	46.76 ± 0.12 ^a^	11.11 ± 0.11 ^a^	0.48 ± 0.05 ^a^
**18**	KWA_W	20.80 ± 0.04 ^b^	3.94 ± 0.00 ^c^	0.13 ± 0.02 ^c^
**19**	YARK_B	47.64 ± 0.04 ^a^	10.40 ± 0.20 ^a^	0.47 ± 0.09 ^a^
**20**	YARK_W	21.29 ± 0.09 ^b^	2.30 ± 0.01 ^c^	0.21 ± 0.06 ^b^
**21**	DANB_B	47.19 ± 0.25 ^a^	10.65 ± 0.31 ^a^	0.48 ± 0.05 ^a^
**22**	DANB_W	10.89 ± 0.09 ^d^	1.54 ± 0.01 ^d^	0.12 ± 0.00 ^c^
**23**	BAD_B	46.16 ± 0.05 ^a^	10.85 ± 0.22 ^a^	0.50 ± 0.04 ^a^
**24**	BAD_W	14.89 ± 0.09 ^c^	0.54 ± 0.01 ^d^	0.10 ± 0.00 ^c^
**25**	FARO_B	46.44 ± 0.02 ^a^	11.39 ± 0.02 ^a^	0.48 ± 0.07 ^a^
**26**	FARO_W	11.68 ± 0.23 ^d^	0.71 ± 0.02 ^d^	0.11 ± 0.00 ^c^
**27**	JIR_B	46.54 ± 0.67 ^a^	10.91 ± 0.24 ^a^	0.49 ± 0.06 ^a^
**28**	JIR_W	18.90 ± 0.87 ^b^	0.47 ± 0.01 ^d^	0.19 ± 0.00 ^b^
**29**	YARM_B	48.15 ± 0.18 ^a^	11.33 ± 0.42 ^a^	0.50 ± 0.08 ^a^
**30**	YARM_W	11.42 ± 0.29 ^c^	0.18 ± 0.01 ^d^	0.20 ± 0.00 ^b^

Mean ± SD values within a column denoted by the same letters in superscript are not significantly different (*p* > 0.05). _B = brown rice, _W = white rice (n = 3). *Akai maza hajj* = AKM, *Baingila* = BAI, *Baburashi* = BAB, *Jamila* = JAM, *Maibakincarki* = MBC, *Yarwasagi* = YARW, *Kwandala* = KWA, *Yarkatabore* = YARK, *Danboto* = DANB, *Bakindanboto* = BAD, *Faro 44* = FARO, *Jirkita* = JIR, *Yarkukuma* = YARM, *Dankaushi* = DAN, *Jeep* = JEEP.

**Table 3 molecules-28-00532-t003:** Body weight changes and negative geotaxis for male and female flies after 7 days of diet intervention.

	Male		Female		
S/N	Rice Cultivars	Weight Changes (mg)	Negative Geotaxis (No of Flies/10 s)	Weight Changes (mg)	Negative Geotaxis (No of Flies/10 s)
**1**	NCD	4.80 ± 0.10 ^a^	9.67 ± 0.08 ^a^	9.40 ± 0.36 ^a^	9.33 ± 0.28 ^a^
**2**	HFD	12.40 ± 0.46 ^c^	6.33 ± 0.58 ^b^	13.20 ± 0.30 ^c^	4.67 ± 0.56 ^b^
**3**	AKM_B	4.23 ± 0.15 ^a^	9.33 ± 0.35 ^a^	8.53 ± 0.12 ^a^	8.67 ± 0.18 ^a^
**4**	AKM_W	8.50 ± 0.30 ^b^	7.03 ± 0.05 ^b^	11.77 ± 0.25 ^b^	6.33 ± 0.50 ^b^
**5**	BAI_B	6.17 ± 0.31 ^a^	8.67 ± 0.53 ^a^	9.93 ± 0.15 ^a^	8.33 ± 0.53 ^a^
**6**	BAI_W	10.50 ± 0.30 ^b^	7.00 ± 0.00 ^b^	11.20 ± 0.30 ^b^	6.00 ± 0.00 ^b^
**7**	BAB_B	4.97 ± 0.21 ^a^	8.67 ± 0.03 ^a^	10.10 ± 0.35 ^a^	8.00 ± 0.00 ^a^
**8**	BAB_W	10.13 ± 0.21 ^b^	7.13 ± 0.25 ^b^	12.57 ± 0.21 ^c^	6.00 ± 0.00 ^b^
**9**	DAN_B	4.80 ± 0.10 ^a^	9.33 ± 0.45 ^a^	10.00 ± 0.36 ^a^	8.67 ± 0.50 ^a^
**10**	DAN_W	9.07 ± 0.42 ^b^	7.23 ± 0.08 ^b^	12.57 ± 0.21 ^c^	6.33 ± 0.45 ^b^
**11**	JEEP_B	5.97 ± 0.55 ^a^	9.00 ± 0.00 ^a^	9.90 ± 0.20 ^a^	8.33 ± 0.58 ^a^
**12**	JEEP_W	11.17 ± 0.85 ^c^	7.67 ± 0.28 ^b^	12.97 ± 0.21 ^c^	6.33 ± 0.36 ^b^
**13**	JAM_B	5.77 ± 0.06 ^a^	8.53 ± 0.50 ^a^	8.50 ± 0.00 ^a^	8.33 ± 0.58 ^a^
**14**	JAM_W	11.40 ± 0.00 ^c^	7.33 ± 0.51 ^b^	11.13 ± 0.06 ^b^	6.67 ± 0.51 ^b^
**15**	MBC_B	5.03 ± 0.12 ^a^	9.00 ± 0.00 ^a^	8.67 ± 0.06 ^a^	9.00 ± 1.00 ^a^
**16**	MBC_W	9.47 ± 0.12 ^b^	6.73 ± 0.54 ^b^	12.63 ± 0.06 ^c^	5.67 ± 0.49 ^b^
**17**	YARW_B	6.00 ± 0.00 ^a^	8.67 ± 0.48 ^a^	8.80 ± 0.17 ^a^	9.67 ± 0.53 ^a^
**18**	YARW_W	9.87 ± 0.06 ^b^	6.33 ± 0.57 ^b^	12.97 ± 0.12 ^c^	5.67 ± 1.15 ^b^
**19**	KWA_B	4.30 ± 0.00 ^a^	9.33 ± 0.39 ^a^	8.30 ± 0.00 ^a^	9.33 ± 0.54 ^a^
**20**	KWA_W	8.83 ± 0.06 ^b^	7.33 ± 0.62 ^b^	11.83 ± 0.06 ^b^	6.47 ± 0.57 ^b^
**21**	YARK_B	5.43 ± 0.12 ^a^	8.00 ± 0.00 ^a^	8.80 ± 0.10 ^a^	8.67 ± 1.15 ^a^
**22**	YARK_W	9.77 ± 0.06 ^b^	6.67 ± 0.40 ^b^	12.77 ± 0.06 ^c^	6.33 ± 0.51 ^b^
**23**	DANB_B	5.57 ± 0.23 ^a^	8.33 ± 0.55 ^a^	9.13 ± 0.23 ^a^	9.33 ± 0.29 ^a^
**24**	DANB_W	12.10 ± 0.10 ^c^	7.33 ± 0.48 ^b^	13.13 ± 0.12 ^c^	5.33 ± 0.52 ^b^
**25**	BAD_B	4.37 ± 0.32 ^a^	8.67 ± 0.33 ^a^	9.27 ± 0.06 ^a^	8.33 ± 0.47 ^a^
**26**	BAD_W	12.27 ± 0.31 ^c^	3.33 ± 0.52 ^c^	14.20 ± 0.10 ^c^	5.00 ± 0.00 ^b^
**27**	FARO_B	6.07 ± 0.06 ^a^	9.00 ± 0.00 ^a^	9.47 ± 0.42 ^a^	8.33 ± 0.58 ^a^
**28**	FARO_W	11.50 ± 0.00 ^c^	5.33 ± 0.25 ^c^	12.03 ± 0.06 ^c^	6.36 ± 0.43 ^b^
**29**	JIR_B	6.10 ± 0.00 ^a^	8.00 ± 0.00 ^a^	8.80 ± 0.00 ^a^	9.00 ± 0.00 ^a^
**30**	JIR_W	11.47 ± 0.06 ^c^	7.67 ± 0.58 ^b^	13.57 ± 0.40 ^c^	6.00 ± 0.00 ^b^
**31**	YARM_B	5.53 ± 0.64 ^a^	9.00 ± 0.00 ^a^	9.00 ± 0.00 ^a^	8.33 ± 0.58 ^a^
**32**	YARM_W	11.20 ± 0.00 ^c^	6.70 ± 0.02 ^b^	13.50 ± 0.00 ^c^	5.00 ± 0.00 ^b^

Mean ± SD values within a column denoted by the same letters in superscript are not significantly different (*p* > 0.05). _B, Brown rice; _W, White rice; (n, 30). NCD = normal diet, HFD = high-fat diet *Akai maza hajj* = AKM, *Baingila* = BAI, *Baburashi* = BAB, *Jamila* = JAM, *Maibakincarki* = MBC, *Yarwasagi* = YARW, *Kwandala* = KWA, *Yarkatabore* = YARK, *Danboto* = DANB, *Bakindanboto* = BAD, *Faro 44* = FARO, *Jirkita* = JIR, *Yarkukuma* = YARM, *Dankaushi* = DAN, *Jeep* = JEEP.

**Table 4 molecules-28-00532-t004:** Gene name, accession number and primer sequences used in the analysis of drosophila melanogaster genes.

S/N	Gene Name	Accession Number	Left	Right
**1.**	*PEPCK*	NM_079060.3	CCTCGATGGCATGAAGGATAAG	GACTCGAATAGGTGCGAATATC
**2.**	*IRS-2*	NM_168448.3	CGGGCGCTTAATACATCACTA	CTGCCGGTCAAATCTCCTATC
**3.**	*ACC*	NM_079288.3	ATCCCGTGATTTCCACACAAG	AGTTATCCTCCTCCTCGAACTC
**4.**	*RPL-32*	NM_079814.3	GTCGTCGCTTTGTCATCT	GCAGGTTGTAGCCCTTCTT

*PEPCK* = phosphoenolpyruvate carboxylate kinase, *IRS-2* = insulin receptor substrate-2, ACC =Acetyl CoA carboxylase1, *RPL-32* = 60S ribosomal protein large subunit-32.

## Data Availability

The data presented in this study are available within the article and Appendix A.

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
