# Peer review of "Nutrigenomic Effects of White Rice and Brown Rice on the Pathogenesis of Metabolic Disorders in a Fruit Fly Model"

_molecules, 2023, doi:10.3390/molecules28020532_

Round 1
Reviewer 1 Report
The study was interesting and the results were clearly discussed.
1. L48, different varieties of brown rice possess different phenolic acids profiles, some of which indicate high level of anthocyanins. Please refer to DOI: 10.3390/foods11111552.
2. 2.3.5, the calculation of Carbohydrate Contents was not appropriate. Some chemical analysis should be carried out.
3. 2.4.1, why the authors prepared Methanolic Extracts, but not ethanolic extracts for total phenolic and flavonoid content analysis.
4. The methods especially for proximate analysis should be condensed and briefly introduced.
5. 3.3, the title should be “Effects on Body Weight Changes”.
Author Response
The study was interesting and the results were clearly discussed.
Response: Thank you for finding time to review our manuscript with commendation
- L48, different varieties of brown rice possess different phenolic acids profiles, some of which indicate high level of anthocyanins. Please refer to DOI: 10.3390/foods11111552.
Response 1: Thank you for sharing with us this information. This has been included and cited.
- 3.5, the calculation of Carbohydrate Contents was not appropriate. Some chemical analysis should be carried out.
Response 2: We adopted the protocol from the standardised official methods of analysis of the Association of Official Analytical Chemist (AOAC), as cited (now 16, 24, and 25).
- 4.1, why the authors prepared Methanolic Extracts, but not ethanolic extracts for total phenolic and flavonoid content analysis.
Response 3: We also adopted the protocol from previously published and similar studies, and the references were cited (now 16 and 24).
- The methods especially for proximate analysis should be condensed and briefly introduced.
Response 4: The method for proximate analysis has been condensed and briefly introduced.
- 3.3, the title should be “Effects on Body Weight Changes”.
Response 5: Thank you for calling our attention to this. It has been corrected.
Reviewer 2 Report
Dear authors,
Thank you for submitting the manuscript: Nutrigenomic Effects of White Rice and Brown Rice on The Pathogenesis of Metabolic Disorders in A Fruit Fly Model to the Molecules.
In the presented study fifteen cultivars of paddy rice that are predominantly consumed in Northwestern Nigeria were analysed for their nutritional composition, bioactive contents and effects on metabolic outcomes in a fruit fly model were explored.
The Introduction part is well written, and used references are novel. Materials and methods section is good explained and logically arranged. Results, as well as discussion part is nicely presented as well as obtained results, and statistically analysed. In the presented manuscript authors came up with a convincing clear conclusions.
I suggest the acceptance of the manuscript after English language corrections.
.
Author Response
Dear authors,
Thank you for submitting the manuscript: Nutrigenomic Effects of White Rice and Brown Rice on The Pathogenesis of Metabolic Disorders in A Fruit Fly Model to the Molecules.
In the presented study fifteen cultivars of paddy rice that are predominantly consumed in Northwestern Nigeria were analysed for their nutritional composition, bioactive contents and effects on metabolic outcomes in a fruit fly model were explored.
The Introduction part is well written, and used references are novel. Materials and methods section is good explained and logically arranged. Results, as well as discussion part is nicely presented as well as obtained results, and statistically analysed. In the presented manuscript authors came up with a convincing clear conclusions.
Response: We really appreciate the reviewer for finding time to review our manuscript and accompanying it with commendations. This is a morale booster for our science.
I suggest the acceptance of the manuscript after English language corrections.
Response: Thank you for finding our manuscript worthy of acceptance, we have made appropriate correction to the English language. The changes have been tracked accordingly in the revised copy.